# Modulate Your Spectrum in Self-Supervised Learning

## Abstract

Whitening loss provides theoretical guarantee in avoiding feature collapse for self-supervised learning (SSL) using joint embedding architectures. One typical implementation of whitening loss is hard whitening that designs whitening transformation over embedding and imposes the loss on the whitened output. In this paper, we propose spectral transformation (ST) framework to map the spectrum of embedding to a desired distribution during forward pass, and to modulate the spectrum of embedding by implicit gradient update during backward pass. We show that whitening transformation is a special instance of ST by definition, and there exist other instances that can avoid collapse by our empirical investigation. Furthermore, we propose a new instance of ST, called IterNorm with trace loss (INTL). We theoretically prove that INTL can avoid collapse and modulate the spectrum of embedding towards an equal-eigenvalue distribution during the course of optimization. Moreover, INTL achieves 76.6% top-1 accuracy in linear evaluation on ImageNet using ResNet-50, which exceeds the performance of the supervised baseline, and this result is obtained by using a batch size of only 256. Comprehensive experiments show that INTL is a promising SSL method in practice.

## 1 Introduction

Self-supervised learning (SSL) via joint embedding architectures to learn visual representations has made significant progress over the last several years [1, 18, 7, 9, 2, 30], almost outperforming their supervised counterpart on many downstream tasks [28, 23, 32]. This paradigm addresses to train a dual pair of networks to produce similar embeddings for different views of the same image [9]. One main challenge with the joint embedding architectures is how to prevent a *collapse* of the representation, in which the two branches ignore the inputs and produce identical and constant outputs [9]. A variety of methods have been proposed to successfully avoid *collapse*, including contrastive learning methods [41, 18, 34] that attract different views from the same image (positive pairs) while pull apart different images (negative pairs), and non-contrastive methods [16, 9] that directly match the positive targets without introducing negative pairs.

The collapse problem is further generalized into *dimensional collapse* [20, 24] (or *informational collapse* [2]), where the embedding vectors only span a lower-dimensional subspace and would be highly correlated. In this case, the covariance matrix of embedding has certain zero eigenvalues, which degenerates the representation in SSL. To prevent *dimensional collapse*, a theoretically motivated paradigm, called whitening loss, is proposed by minimizing the distance between embeddings of positive pairs under the condition that embeddings from different views are whitened [12, 20]. One typical implementation of whitening loss is hard whitening [12, 40] that designs whitening transformation over mini-batch data and imposes the loss on the whitened output [12, 20, 40]. We note that the whitening transformation is a function over embedding during forward pass, and modulates the spectrum of embedding implicitly during backward pass when minimizing the objective. This raises questions whether there exist other functions over embedding can avoid collapse? If yes, how the function affects the spectrum of embedding?

This paper proposes spectral transformation (ST), a framework to modulate the spectrum of embedding in joint embedding architecture. ST maps the spectrum of embedding to a desired distribution during forward pass, and modulates the spectrum of embedding by implicit gradient update during backward pass (Figure 1). This framework provides a way to seek for functions beyond whitening transformation that can avoid dimensional collapse. We show that whitening transformation is a special instance of ST

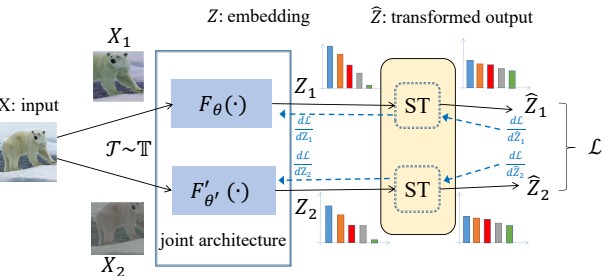

Figure 1: The framework using spectral transformation (ST) to modulate the spectrum of embedding in joint embedding architecture for SSL.

using a power function by definition, and there exist other power functions that can avoid dimensional collapse by our empirical investigation (see Section 3.2 for details). We demonstrate that IterNorm [22], an approximating whitening method by using Newton's iterations [3, 42], is also an instance of ST, and show that IterNorm with different iteration number corresponds to different ST (see Section 3.3 for details). We further theoretically characterize how the spectrum evolves as the increasing of iteration number of IterNorm.

We empirically observe that IterNorm suffers from severe dimensional collapse and mostly fails to train the model in SSL unexpectedly, unlike its benefits in approximating whitening for supervised learning [22]. We thus propose IterNorm with trace loss (INTL), a simple solution to address the failure of IterNorm, by adding an extra penalty on the transformed output. Moreover, we theoretically demonstrate that INTL can avoid dimensional collapse, and reveal its mechanism in encouraging the covariance matrix of embedding to have equal eigenvalues. We conduct comprehensive experiments and show that INTL is a promising SSL method in practice. *E.g.*, INTL achieves 76.6% top-1 accuracy in linear evaluation on ImageNet using ResNet-50, which exceeds the performance of the supervised baseline, and this result is obtained by using a batch size of only 256. Our main contributions are summarized as follows:

- We propose spectral transformation, a framework to modulate the spectrum of embedding and to seek for functions beyond whitening that can avoid dimensional collapse. We show there exist other functions that can avoid dimensional collapse by empirical observations.

- We propose a new instance of ST, called IterNorm with trace loss (INTL). We theoretically prove that INTL can avoid collapse and modulate the spectrum of embedding towards an equal-eigenvalue distribution during the course of optimization.

- INTL is a promising SSL method in practice. INTL is on par with or outperforms the state-of-the-art SSL methods on standard benchmarks. Furthermore, these results are obtained by using a relatively small batch size.

## 2 Related Work

Our work is related to the SSL methods that address the feature collapse problem when using joint embedding architectures. Contrastive learning prevents collapse by attracting positive samples closer, and spreading negative samples apart [41, 43]. In these methods, negative samples play an important role and need to be well designed [29, 1, 19]. MoCos [18, 8] builds a memory bank with a momentum encoder to provide consistent negative samples, while SimCLR [7] addresses that more negative samples in a batch with strong data augmentations perform better. Our proposed INTL can avoid collapse and work well without negative samples.

Non-contrastive methods by designing asymmetric network architecture avoid feature collapse without introducing negative pairs explicitly [4, 5, 26, 16, 9]. BYOL [16] appends a predictor after the online network and introduce momentum into the target network. SimSiam [9] further generalizes BYOL by empirically showing that stop-gradient is essential for preventing trivial solutions. Other progresses include a cluster assignment prediction using Sinkhorn-Knopp algorithm [5], and an asymmetric pipeline with a self-distillation loss for Vision Transformers [6]. It remains not clear how the asymmetric network avoids collapse without negative pairs, leaving the debates on batch normalization (BN) [13, 39, 33] and stop-gradient [9, 45], even though preliminary works have attempted to analyze the training dynamics [38] and build a connection between non-contrastive and contrastive methods [36, 14]. Our work addresses the more challenging dimensional collapse problem, and theoretically shows that our INTL can avoid dimensional collapse.

Whitening loss is a theoretically motivated paradigm to prevent dimensional collapse [12]. One typical implementation of whitening loss is hard whitening that designs whitening transformation over mini-batch data and imposes the loss on the whitened output. The designed whitening transformation includes batch whitening in W-MSE [12] and Shuffled-DBN [20], channel whitening in CW-RGP [40], and the combination of both in Zero-CL [47]. Our proposed ST generalizes whitening transformation and provides a frame to modulate the spectrum of embedding. Our proposed INTL can improve these work in training stability and performance, by replacing whitening transformation with IterNorm [22] and imposing an additional trace loss on the transformed output. Furthermore, we theoretically show that our proposed INTL encourages the covariance matrix of embedding having equal eigenvalues.

Another way to implement whitening loss is soft whitening that imposes a whitening penalty as regularization on the embedding, including Barlow Twins [44], VICReg [2] and CCA-SSG [46]. Different from these works, our proposed INTL imposes the trace loss on the approximated whitened output, which implicitly encourages the covariance matrix of embedding having equal eigenvalues to avoid dimensional collapse.

There are also theoretical works analyzing how dimensional collapse occurs [20, 24] and how it can be avoided by using whitening loss [20, 40]. The recent works [17, 15] further discuss how to characterize the magnitude of dimensional collapse, and connect the spectrum of a representation to a power law. They show the coefficient of the power law is a strong indicator for the effects of the representation. Different from these works, our theoretical analysis presents a new thought in demonstrating how to avoid dimensional collapse, which provides theoretical basis for our proposed INTL.

# 3 Spectral Transformation beyond Whitening

## 3.1 Preliminary and Notation

**Joint embedding architectures.** Let $\mathbf{x}$ denote the input sampled uniformly from a set of images $\mathbb{D}$, and $\mathbb{T}$ denote the set of data transformations available for augmentation. We consider a pair of neural networks $F_\theta$ and $F'_{\theta'}$, parameterized by $\theta$ and $\theta'$ respectively. They take as input two randomly augmented views, $\mathbf{x}^{(1)} = \mathcal{T}_1(\mathbf{x})$ and $\mathbf{x}^{(2)} = \mathcal{T}_2(\mathbf{x})$, where $\mathcal{T}_{1,2} \in \mathbb{T}$; and they output the *embedding* $\mathbf{z}^{(1)} = F_\theta(\mathbf{x}^{(1)})$ and $\mathbf{z}^{(2)} = F'_{\theta'}(\mathbf{x}^{(2)})$. The networks are trained with an objective function that minimizes the distance between embeddings obtained from different views of the same image:

$$\mathcal{L}(\mathbf{x}, \theta) = \mathbb{E}_{\mathbf{x} \sim \mathbb{D}, \, \mathcal{T}_{1,2} \sim \mathbb{T}} \; \ell\big(F_\theta(\mathcal{T}_1(\mathbf{x})), F'_{\theta'}(\mathcal{T}_2(\mathbf{x}))\big). \tag{1}$$

where $\ell(\cdot, \cdot)$ is a loss function. The mean square error (MSE) of $L_2-$normalized vectors as $\ell(\mathbf{z}^{(1)}, \mathbf{z}^{(2)}) = \|\frac{\mathbf{z}^{(1)}}{\|\mathbf{z}^{(1)}\|_2} - \frac{\mathbf{z}^{(2)}}{\|\mathbf{z}^{(2)}\|_2}\|_2^2$ is usually used as the loss function [9]. This loss is also equivalent to the negative cosine similarity, up to a scale of $\frac{1}{2}$ and an optimization irrelevant constant [9]. This architecture is also called *Siamese Network* [9], if $F_\theta = F'_{\theta'}$. Another variant distinguishes the networks into target network $F'_{\theta'}$ and online network $F_\theta$, and updates the weight $\theta'$ of target network through exponential moving average (EMA) [8, 16] over $\theta$ of online network.

**Feature collapse.** While minimizing Eqn. 1, a trivial solution known as *complete collapse* could occur such that $F_\theta(\mathbf{x}) \equiv \mathbf{c}, \forall \mathbf{x} \in \mathbb{D}$. Moreover, a weaker collapse condition called *dimensional collapse* can be easily arrived, for which the projected features collapse into a low-dimensional manifold. To express dimensional collapse more mathematically, we refer to dimensional collapse as the phenomenon that one or certain eigenvalues of the covariance matrix of feature vectors degenerate to 0. Therefore, we can determine the occurrence of dimensional collapse by observing the spectrum of the covariance matrix.

**Whitening loss.** To address the collapse problem, whitening loss [12] is proposed to minimize Eqn. 1, under the condition that *embeddings* from different views are whitened. Whitening loss provides theoretical guarantee in avoiding (dimensional) collapse, since the embedding is whitened with all axes decorrelated [12, 20]. Ermolov *et al.* [12] propose to whiten the mini-batch embedding $\mathbf{Z} \in \mathbb{R}^{d \times m}$ using batch whitening (BW) [21, 35] and impose the loss on the whitened output $\widehat{\mathbf{Z}} \in \mathbb{R}^{d \times m}$, given the mini-batch inputs $\mathbf{X}$ with size of $m$, as follows:

$$\min_\theta \mathcal{L}(\mathbf{X}; \theta) = \mathbb{E}_{\mathbf{X} \sim \mathbb{D}, \, \mathcal{T}_{1,2} \sim \mathbb{T}} \|\widehat{\mathbf{Z}}^{(1)} - \widehat{\mathbf{Z}}^{(2)}\|_F^2$$
$$with \; \widehat{\mathbf{Z}}^{(v)} = \Sigma^{-\frac{1}{2}} \mathbf{Z}^{(v)}, \, v \in \{1, 2\}, \tag{2}$$

where $\Sigma = \frac{1}{m}\mathbf{Z}\mathbf{Z}^T$ is the covariance matrix of embedding[1]. $\Sigma^{-\frac{1}{2}}$ is called the whitening matrix, and is calculated either by Cholesky decomposition in [12] or by eigen-decomposition in [20]. *E.g.*, zero-phase component analysis (ZCA) whitening [21] calculates $\Sigma^{-\frac{1}{2}} = \mathbf{U}\Lambda^{-\frac{1}{2}}\mathbf{U}^T$, where $\Lambda = \mathrm{diag}(\lambda_1,\ldots,\lambda_d)$ and $\mathbf{U} = [\mathbf{u}_1,...,\mathbf{u}_d]$ are the eigenvalues and associated eigenvectors of $\Sigma$, *i.e.*, $\mathbf{U}\Lambda\mathbf{U}^T = \Sigma$. One intriguing result shown in [40] is that hard whitening can avoid collapse by only constraining the embedding $\mathbf{Z}$ to be full-rank, but not whitened.

We note that the whitening transformation is a function over embedding $\mathbf{Z}$ during forward pass, and modulates the spectrum of embedding $\mathbf{Z}$ implicitly during backward pass when minimizing MSE loss imposed on the whitened output. This raises questions whether there are other functions over embedding $\mathbf{Z}$ can avoid collapse? If yes, how the function affects the spectrum of embedding $\mathbf{Z}$?

## 3.2 Spectral Transformation

In this section, we extend the whitening transformation to spectral transformation, a more general view to characterize the modulation on the spectrum of embedding, and empirically investigate the interaction between the spectrum of the covariance matrix of $\widehat{\mathbf{Z}}$ and collapse of the SSL model.

**Definition 1.** *(**Spectral Transformation**) Given any one-variable mapping function $g(\cdot)$ in the definition domain $\lambda(\mathbf{Z}) = \{\lambda_1, \lambda_2, \ldots, \lambda_d\}$, spectral transformation (ST) maps the spectrum $\lambda(\mathbf{Z})$ to $g(\lambda(\mathbf{Z})) = \{g(\lambda_1), g(\lambda_2), \ldots, g(\lambda_d)\}$. Accordingly, for a $d \times d$ real symmetric matrix $\Sigma$, spectral transformation $g(\cdot)$ on $\Sigma = \sum_{i=1}^{d} \lambda_i \mathbf{u}_i \mathbf{u}_i^T$ is defined as $g(\Sigma) = \sum_{i=1}^{d} g(\lambda_i)\mathbf{u}_i\mathbf{u}_i^T$. We denote $g(\Lambda) = diag(g(\lambda(\mathbf{Z})))$, and $\Phi_{ST} = g(\Sigma) = \mathbf{U}g(\Lambda)\mathbf{U}^T$ is the transformation matrix.*

The output of spectral transformation is calculated by $\widehat{\mathbf{Z}} = \Phi_{ST}\mathbf{Z} = \mathbf{U}g(\Lambda)\mathbf{U}^T\mathbf{Z}$. The covariance matrix of $\widehat{\mathbf{Z}}$ is:

$$\Sigma_{\widehat{\mathbf{Z}}} = \frac{1}{m}\widehat{\mathbf{Z}}\widehat{\mathbf{Z}}^T = \mathbf{U}\Lambda g^2(\Lambda)\mathbf{U}^T.$$

Based on this formula, the essence of spectral transformation is mapping the spectrum $\lambda(\mathbf{Z}) = \{\lambda_1, \lambda_2, \ldots, \lambda_d\}$ to $\lambda(\widehat{\mathbf{Z}}) = \{\lambda_1 g^2(\lambda_1), \lambda_2 g^2(\lambda_2), \ldots, \lambda_d g^2(\lambda_d)\}$.

**ST using power functions.** Whitening is a special instance of spectral transformation, where $g(\cdot)$ is a power function $g(\lambda) = \lambda^{-\frac{1}{2}}$. We further study the mechanism of this power transformation, where we consider a more general transformation $g(\lambda) = \lambda^{-p}, p \in (-\infty, +\infty)$ for ST. Based on Definition 1, this general power transformation is mapping the spectrum $\lambda(\mathbf{Z})$ to $\lambda(\widehat{\mathbf{Z}}) = \{\lambda_1^{1-2p}, \lambda_2^{1-2p}, \ldots, \lambda_d^{1-2p}\}$, *e.g.*, $\lambda(\widehat{\mathbf{Z}}) = \{1, 1, \ldots, 1\}$ when using whitening with $p = \frac{1}{2}$.

We first conduct experiments on the 2D dataset with varying $p$ and visualize outputs of the toy models in Figure 2(a). We observe the toy model seems to perform well to avoid collapse although the trans-

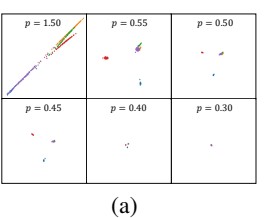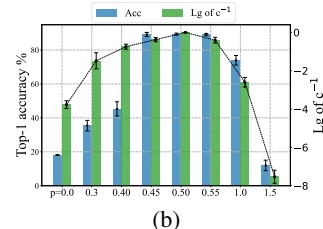

(a)  (b)

Figure 2: Investigate ST using power functions. We choose several $p$ from 0 to 1.5. We show (a) the visualization of the toy model output; (b) top-1 accuracy and condition indicator (we use the inverse of the condition number $c^{-1} = \frac{\lambda_d}{\lambda_1}$) on CIFAR-10 with ResNet-18. The results on CIFAR-10 are averaged over five runs, with standard deviation shown as error bars and we show the details of experimental setup in *supplementary materials*. Similar phenomena can be observed when using other datasets (*e.g.*, ImageNet) and other networks (*e.g.*, ResNet-50).

formed output is not ideally whitened, when $p$ is in the neighborhood of $0.5$, e.g. $0.45 \sim 0.55$. But when $p$ gradually deviates from $0.5$, collapse occurs. We then conduct experiments on real-world datasets to confirm these phenomena. The results shown in Figure 2(b) are consistent with the above phenomena. When $p$ is set to $0.45$ or $0.55$, the model remains high evaluation performance as the one of $p = 0.5$. When $p$ is in the neighborhood of $0.5$, the transformed output has a well-conditioned spectrum that each eigenvalue approaches 1. When $p$ deviates from $0.5$ to a certain extent, the

---

[1]The embedding is usually centralized by performing $\mathbf{Z} := \mathbf{Z}(\mathbf{I} - \frac{1}{m}\mathbf{1} \cdot \mathbf{1}^T)$ for whitening, and we assume $\mathbf{Z}$ is centralized in this paper for simplifying discussion.

spectrum of the transformed output is not well-conditioned, which is closely related to collapse of the embedding. Therefore, we empirically show that if a certain ST could obtain a well-conditioned spectrum of transformed output, collapse could be avoided.

Based on the above results, we empirically observe that the spectral transformation $g(\lambda) = \lambda^{-p}$ with $p$ around 0.5 can avoid collapse. Therefore, we can design new algorithms based on our framework to avoid collapse.

## 3.3 Implicit Spectral Transformation using Newton's Iteration

One problem of ST using power functions $g(\lambda) = \lambda^{-p}$ ($p$ is around 0.5) is the numerical instability, when calculating eigenvalues $\lambda$ and eigenvectors $\mathbf{U}$ using eigen-decomposition if the covariance matrix is ill-conditioned [31]. We provide detailed experiments and analysis in *supplementary materials* to confirm the existence of this problem in SSL.

Naturally, if we can implement a spectral transformation that can modulate the spectrum without explicitly calculating $\lambda$ or $\mathbf{U}$, this problem can be solved. Indeed, we note an approximate whitening method by using Newton's iteration, called iterative normalization (IterNorm) [22], is proposed to address the numerical problem of batch whitening in supervised learning. Specifically, given the centralized embedding $\mathbf{Z}$, iteration number $T$ and the trace-normalized covariance matrix $\Sigma_N = \Sigma/tr(\Sigma)$, it performs Newton's iteration as follows.

$$\begin{cases} \mathbf{P}_0 = \mathbf{I} \\ \mathbf{P}_k = \frac{1}{2}(3\mathbf{P}_{k-1} - \mathbf{P}_{k-1}^3\Sigma_N), \;\; k = 1, 2, ..., T. \end{cases} \tag{3}$$

The whitening matrix $\Sigma^{-\frac{1}{2}}$ is approximated by $\Phi_T = \mathbf{P}_T/\sqrt{tr(\Sigma)}$ and we have the whitened output $\widehat{\mathbf{Z}} = \Phi_T \mathbf{Z}$. When $T \to +\infty$, $\Phi_T \to \Sigma^{-\frac{1}{2}}$ and the covariance matrix of $\widehat{\mathbf{Z}}$ will be an identity matrix. Here, we theoretically show that IterNorm is also an instance of spectral transformation as follows.

**Theorem 1.** *Define one-variable iterative function $f_T(x)$, satisfying*

$$f_{k+1}(x) = \frac{3}{2}f_k(x) - \frac{1}{2}x{f_k}^3(x), k \geq 0; \; f_0(x) = 1.$$

*The mapping function of IterNorm is*

$$g(\lambda) = f_T(\tfrac{\lambda}{tr(\Sigma)})/\sqrt{tr(\Sigma)},$$

*Without calculating $\lambda$ or $\mathbf{U}$, IterNorm implicitly maps $\forall \lambda_i \in \lambda(\mathbf{Z})$ to $\widehat{\lambda}_i = \frac{\lambda_i}{tr(\Sigma)}{f_T}^2(\frac{\lambda_i}{tr(\Sigma)})$.*

The proof is shown in *supplementary materials*. For simplicity, we define the T-whitening function of IterNorm $h_T(x) = x{f_T}^2(x)$, which obtains the spectrum of transformed output. Based on the fact that the covariance matrix of transformed output will be identity when $T$ of IterNorm increases to infinity [3], we thus have

$$\forall \lambda_i > 0, \lim_{T \to \infty} h_T(\frac{\lambda_i}{tr(\Sigma)}) = 1. \tag{4}$$

Different iteration numbers $T$ of IterNorm imply different T-whitening functions $h_T(\cdot)$. It is interesting to analyze the characteristics of $h_T(\cdot)$.

**Proposition 1.** *Given $x \in (0, 1)$, $\forall T \in \mathbb{N}$ we have $h_T(x) \in (0, 1)$ and $h'_T(x) > 0$.*

The proof of Proposition 1 is shown in *supplementary materials*. Proposition 1 states $h_T(x)$ is a monotone increasing function for $x \in (0, 1)$ and its range is also in $(0, 1)$. Since $\frac{\lambda_i}{tr(\Sigma)} \in (0, 1)$, $\forall \lambda_i > 0$, we have

$$\forall T \in \mathbb{N}, \lambda_i > \lambda_j > 0 \Longrightarrow 1 > \widehat{\lambda}_i > \widehat{\lambda}_j > 0. \tag{5}$$

Formula 5 indicates that IterNorm maps all non-zero eigenvalues to $(0, 1)$ and preserves monotonicity.

**Proposition 2.** *Given $x \in (0, 1)$, $\forall T \in \mathbb{N}$, we have $h_{T+1}(x) > h_T(x)$.*

The proof of Proposition 2 is shown in *supplementary materials*. Proposition 2 indicates that IterNorm gradually stretches the eigenvalues towards one as the iteration number $T$ increases. This property of IterNorm theoretically shows that the spectrum of $\widehat{\mathbf{Z}}$ will have better condition if we use a larger iteration number $T$ of IterNorm.

In summary, our analyses theoretically show that IterNorm gradually stretches the eigenvalues towards one as the iteration number $T$ increases, and the smaller the eigenvalue is, the larger $T$ is required to approach one.

# 4  Iterative Normalization with Trace Loss

It is expected that IterNorm, as a kind of spectral transformation, can avoid collapse and obtain good performance in SSL, due to its benefits in approximating whitening for supervised learning [22]. However, we empirically observe that IterNorm suffers severe dimensional collapse and mostly fails to train the model in SSL (we postpone the details in Section 4.2.). Based on the analyses in Section 3.2 and 3.3, we propose a simple solution by adding an extra penalty named *trace loss* on the transformed output $\widehat{\mathbf{Z}}$ by IterNorm to ensure a well-conditioned spectrum. It is clear that the sum of eigenvalues of $\Sigma_{\widehat{\mathbf{Z}}}$ is less than or equal to $d$, we thus propose a trace loss that encourages the trace of $\Sigma_{\widehat{\mathbf{Z}}}$ to be its maximum $d$, when $d \le m$. In particular, we design a new method called *IterNorm with trace loss* (INTL) for optimizing the SSL model as[2]:

$$\min_{\theta \in \Theta} \quad INTL(\mathbf{Z}) = \sum_{j=1}^{d}(1 - (\Sigma_{\widehat{\mathbf{Z}}})_{jj})^2, \tag{6}$$

where $\mathbf{Z} = F_\theta(\cdot)$ and $\widehat{\mathbf{Z}} = IterNorm(\mathbf{Z})$. Eqn. 6 can be viewed as an optimization problem over $\theta$ to encourage the trace of $\widehat{\mathbf{Z}}$ to be $d$.

## 4.1  Theoretical Analysis

In this section, we theoretically prove that INTL can avoid collapse, and INTL modulates the spectrum of embedding towards an equal-eigenvalue distribution during the course of optimization.

Note that $\Sigma_{\widehat{\mathbf{Z}}}$ can be expressed using the T-whitening function $h_T(\cdot)$ as $\Sigma_{\widehat{\mathbf{Z}}} = \sum_{i=1}^{d} h_T(x_i)\mathbf{u}_i\mathbf{u}_i^T$, where $x_i = \lambda_i/tr(\Sigma) \ge 0$ and $\sum_{i=1}^{d} x_i = 1$. When the range of $F_\theta(\cdot)$ is wide enough, the optimization problem over $\theta$ (Eqn. 6) can be transformed as the following optimization problem over $\mathbf{x}$ (Eqn. 7) without changing the optimal value (please see *supplementary materials* for the details of derivation):

$$\begin{cases} \min_{\mathbf{x}} \quad INTL(\mathbf{x}) = \sum_{j=1}^{d}\left(\sum_{i=1}^{d}[1 - h_T(x_i)]u_{ji}^2\right)^2 \\ s.t. \quad \sum_{i=1}^{d} x_i = 1 \\ \quad\quad x_i \ge 0, i = 1, \cdots, d, \end{cases} \tag{7}$$

where $u_{ji}$ is the $j$-th elements of vector $\mathbf{u}_i$. In this formulation, we can prove that our proposed INTL can theoretically avoid collapse, as long as the iteration number $T$ of IterNorm is larger than zero.

**Theorem 2.** *Let $\mathbf{x} \in [0,1]^d$, $\forall T \in \mathbb{N}_+$, $INTL(\mathbf{x})$ shown in Eqn. 7 is a strictly convex function. $\mathbf{x}^* = [\frac{1}{d}, \cdots, \frac{1}{d}]^T$ is the unique minimum point as well as the optimal solution to $INTL(\mathbf{x})$.*

The proof is shown in *supplementary materials*. Based on Theorem 2, INTL promotes the equality of all eigenvalues of the covariance matrix of embedding $\mathbf{Z}$ during the course of optimization, which provides a theoretical guarantee to avoid dimensional collapse.

**Connection to hard whitening.**  Hard whitening methods, like W-MSE [12] and shuffle-DBN [20], design a whitening transformation over each view and minimize the distances between the whitened outputs from different views. This mechanism encourages the covariance matrix of *embedding* to be full-rank [40]. Our INTL designs an approximated whitening transformation using IterNorm and imposes an additional trace loss penalty on the (approximately) whitened output. This encourages the covariance matrix of *embedding* having equal eigenvalues.

**Connection to soft whitening.**  Soft whitening methods, like Barlow-Twins [44] and VICReg [2] directly impose a whitening penalty as a regularization on the *embedding*. This encourages the covariance matrix of the *embedding* to be identity (with a **fixed** scalar $\gamma$, *e.g.*, $\gamma\mathbf{I}$). Our INTL imposes

---

[2]Without losing validity, we ignore the MSE term for simplifying discussion.

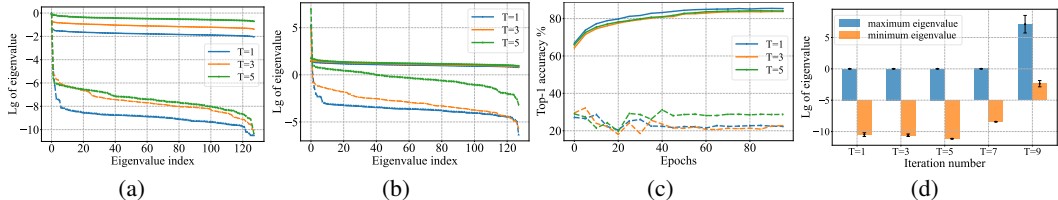

Figure 3: Investigate the effectiveness of IterNorm with and without trace loss. We train the models on CIFAR-10 with ResNet-18 for 100 epochs (details of experimental setup are shown in *supplementary materials*). We apply IterNorm with various iteration numbers $T$, and show the results with (solid lines) and without (dashed lines) trace loss respectively. (a) The spectrum of the transformed output $\widehat{\mathbf{Z}}$; (b) The spectrum of the embedding $\mathbf{Z}$; (c) The top-1 accuracy. (d) indicates that IterNorm (without trace loss) suffers from numeric divergence when using a large iteration number, e.g. $T = 9$. It is noteworthy that when $T \geq 11$, the loss values are all **NAN**, making the model unable to be trained. Similar phenomena can be observed when using other datasets (*e.g.*, ImageNet) and other networks (*e.g.*, ResNet-50).

the penalty on the transformed output, but can be viewed as implicitly encouraging the covariance matrix of the *embedding* to be identity with a **free** scalar (*i.e.*, having equal eigenvalues).

Intuitively, INTL provides the equal-eigenvalues constraint on the covariance matrix of *embedding*, which is a stronger constraint than hard whitening (the full-rank constraint), but a weaker constraint than soft whitening (the whitening constraint). This preliminary but new comparison provides a new way to understand the whitening loss in SSL.

### 4.2 Empirical Analysis

In this section, we empirically show that IterNorm-only fails to avoid collapse, but IterNorm with trace loss can well avoid collapse.

**IterNorm fails to avoid collapse.** In theory, IterNorm can map all non-zero eigenvalues to approach one, with a large enough $T$. In practice, it usually uses a fixed $T$, and it is very likely to encounter small eigenvalues during training. In this case, IterNorm cannot ensure the transformed output has a well-conditioned spectrum (Figure 3(a)), which potentially results in dimensional collapse. One may use a large $T$, however, IterNorm will encounter numeric divergence upon further increasing the iteration number $T$, even though it has converged. *E.g.*, IterNorm suffers from numeric divergence in Figure 3(d) when using $T = 9$, since the maximum eigenvalue of whitened output is around $10^7$, significantly large than 1 (we attribute to the numeric divergence, since this result goes against Proposition 1 and 2, and we further validate it by monitoring the transformed output). It is noteworthy that when $T \geq 11$, the loss values are all **NAN**, making the model unable to be trained. These problems make IterNorm difficult to avoid dimensional collapse in practice.

**The magic of trace loss for IterNorm.** IterNorm with trace loss works significantly different from IterNorm-only. Our experimental results (Figure 3(b)) empirically show that INTL avoid dimensional collapse, which is consistent with our Theorem 2. INTL encourages the equality of all eigenvalues of the covariance matrix of embedding $\mathbf{Z}$ and achieve good evaluation performance (Figure 3(c)) even when the iteration number $T$ is only 1. This equal-eigenvalues optimal solution is derived from the combination of *IterNorm* and *trace loss*. Benefiting from trace loss, IterNorm can obtain well-conditioned spectra for the transformed output during training (Figure 3(a)), which helps to avoid collapse.

## 5 Experiments on Standard SSL Benchmark

In this section, we conduct experiments on standard SSL benchmarks to validate the effectiveness of our proposed INTL. We first evaluate the performance of INTL for classification on ImageNet [11], CIFAR-10/100 [25] and ImageNet-100 [37]. Then we evaluate the effectiveness in transfer learning, for a pre-trained model using INTL. We provide details of implementation and training protocol as well as computational overhead and the full PyTorch-style algorithm in *supplementary materials*.

### 5.1 Evaluation for Classification

**Evaluation on ImageNet.** We train our INTL using ResNet-50 backbone and evaluate the performance using the common linear evaluation protocol on ImageNet. The results are shown in Table 1. Our INTL achieves a top-1 accuracy of $76.6\%$ that exceeds the performance of the supervised baseline [7] and other SSL methods. Moreover, this result is obtained under a batch size of only 256,

Table 1: Evaluation on ImageNet. All results are based on ResNet-50 backbone: (1) linear classification on top of the frozen representations from ImageNet; (2) semi-supervised classification on top of the fine-tuned representations from 1% and 10% of ImageNet samples. Following MoCo [8] and SimSiam [9], our INTL is trained for 800 epochs with a batch size of 256 on 2 A100-40GB GPUs.

| Method | Linear | | Semi-supervised | | | |
| --- | --- | --- | --- | --- | --- | --- |
| | top-1 | top-5 | top-1 | | top-5 | |
| | | | 1% | 10% | 1% | 10% |
| Supervised | 76.5 | - | 25.4 | 56.4 | 48.4 | 80.4 |
| SimCLR [7] | 69.3 | 89.0 | 48.3 | 65.6 | 75.5 | 87.8 |
| MoCo v2 [8] | 71.1 | - | | - | | |
| BYOL [16] | 74.3 | 91.6 | 53.2 | 68.8 | 78.4 | 89.0 |
| SwAV [5] | 75.3 | - | 53.9 | 70.2 | 78.5 | 89.9 |
| SimSiam [9] | 71.3 | - | | - | | |
| W-MSE [12] | 72.6 | - | | - | | |
| DINO [6] | 75.3 | - | | - | | |
| Barlow Twins [44] | 73.2 | 91.0 | 55.0 | 69.7 | 79.2 | 89.3 |
| VICReg [2] | 73.2 | 91.1 | 54.8 | 69.5 | 79.4 | 89.5 |
| **INTL (ours)** | **76.6** | **93.1** | **61.7** | **72.0** | **84.6** | **90.9** |

Table 2: Classification accuracy (top 1 and top-5) of a linear classifier and a 5-nearest neighbors classifier for different loss functions and datasets. The table is mostly inherited from [10]. All methods are based on ResNet-18 with two views and are trained for 1000-epoch on CIFAR-10/100 with a batch size of 256 and 400-epoch on ImageNet-100 with a batch size of 128.

| Method | CIFAR-10 | | | CIFAR-100 | | | ImageNet-100 | | |
| --- | --- | --- | --- | --- | --- | --- | --- | --- | --- |
| | top-1 | 5-nn | top-5 | top-1 | 5-nn | top-5 | top-1 | 5-nn | top-5 |
| SimCLR [7] | 90.74 | 85.13 | 99.75 | 65.78 | 53.19 | 89.04 | 77.64 | 65.78 | 94.06 |
| MoCo V2 [8] | **92.94** | 88.95 | 99.79 | 69.89 | 58.09 | 91.65 | 79.28 | 70.46 | 95.18 |
| BYOL [16] | 92.58 | 87.40 | 99.79 | 70.46 | 56.46 | 91.96 | 80.32 | 68.94 | 94.94 |
| SwAV [5] | 89.17 | 84.18 | 99.68 | 64.88 | 53.32 | 88.78 | 74.28 | 63.84 | 92.84 |
| SimSiam [9] | 90.51 | 86.82 | 99.72 | 66.04 | 55.79 | 89.62 | 78.72 | 67.92 | 94.78 |
| W-MSE [12] | 88.67 | 84.95 | 99.68 | 61.33 | 49.65 | 87.26 | 69.06 | 58.44 | 91.22 |
| DINO [6] | 89.52 | 86.13 | 99.71 | 66.76 | 56.24 | 90.34 | 74.92 | 64.30 | 92.78 |
| Barlow Twins [44] | 92.10 | 88.09 | 99.73 | **70.90** | 59.40 | 91.91 | 80.16 | 72.14 | 95.14 |
| VICReg [2] | 92.07 | 87.38 | 99.74 | 68.54 | 56.32 | 90.83 | 79.40 | 71.94 | 95.02 |
| **INTL (ours)** | 92.60 | **90.03** | **99.80** | 70.88 | **61.90** | **92.13** | **81.68** | **73.46** | **95.42** |

achieving one goal of SSL community that seeks for obtaining good performance with a small batch size.

**Semi-supervised training on ImageNet.** Furthermore, we fine-tune our pre-trained INTL model on a subset of ImageNet. We use subsets of size 1% and 10% using the same split as SimCLR. The semi-supervised results obtained on the ImageNet validation set are also reported in Table 1. It shows that INTL outperforms other baselines with a significant margin.

**Evaluation on small and medium size datasets.** In order to further test the generality of INTL, we also provide the linear evaluation results of INTL on CIFAR-10/100 [25] and ImageNet-100 [37] with ResNet-18 as the backbone. We strictly follow the experimental settings in solo-learn [10] for these datasets. As shown in Table 2, INTL achieves a top-1 accuracy of $92.60\%$ on CIFAR-10, $70.88\%$ on CIFAR-100 and $81.68\%$ on ImageNet-100 which is on par with or exceeds the state-of-the-art methods reproduced by solo-learn. Meanwhile, INTL outperforms other baselines with a significant margin when using 5-nearest neighbors classifier which also indicates that INTL has learned good representations.

## 5.2 Transfer to Downstream Tasks

We examine the representation quality by transferring our pre-trained model to other tasks, including COCO [27] object detection and instance segmentation. We use the baseline of the detection codebase from MoCo [18] for INTL. The results of baselines shown in Table 3 are mostly inherited from [9]. We observe that INTL performs much better than other state-of-the-art approaches on COCO object detection and instance segmentation, which shows the great potential of INTL in transferring to downstream tasks.

Table 3: Transfer Learning. All competitive unsupervised methods are based on 200-epoch pre-training on ImageNet (IN). The table are mostly inherited from [9]. Our INTL is performed with 3 random seeds, with mean and standard deviation reported.

| Method | COCO detection | | | COCO instance seg. | | |
|---|---|---|---|---|---|---|
| | $AP_{50}$ | AP | $AP_{75}$ | $AP_{50}$ | AP | $AP_{75}$ |
| Scratch | 44.0 | 26.4 | 27.8 | 46.9 | 29.3 | 30.8 |
| Supervised | 58.2 | 38.2 | 41.2 | 54.7 | 33.3 | 35.2 |
| SimCLR [7] | 57.7 | 37.9 | 40.9 | 54.6 | 33.3 | 35.3 |
| MoCo v2 [8] | 58.8 | 39.2 | 42.5 | 55.5 | 34.3 | 36.6 |
| BYOL [16] | 57.8 | 37.9 | 40.9 | 54.3 | 33.2 | 35.0 |
| SwAV [5] (repro.) | 60.2 | 39.8 | 43.0 | 56.6 | 34.6 | 36.8 |
| SimSiam [9] | 57.5 | 37.9 | 40.9 | 54.2 | 33.2 | 35.2 |
| W-MSE [12] (repro.) | 60.1 | 39.2 | 42.8 | 56.8 | 34.8 | 36.7 |
| Barlow Twins [44] | 59.0 | 39.2 | 42.5 | 56.0 | 34.3 | 36.5 |
| **INTL (ours)** | $\mathbf{61.2}_{\pm 0.08}$ | $\mathbf{41.2}_{\pm 0.12}$ | $\mathbf{44.7}_{\pm 0.19}$ | $\mathbf{57.8}_{\pm 0.04}$ | $\mathbf{35.7}_{\pm 0.04}$ | $\mathbf{38.1}_{\pm 0.12}$ |

## 5.3 Ablation Study

**Batch size.** Most SSL methods, including certain whitening-based methods, are known to be sensitive to batch sizes, e.g. SimCLR [7], SwAV [5] and W-MSE [12] all require a large batch size (e.g. 4096) to work well. We then test the robustness of INTL to batch sizes. We train INTL on ImageNet for 100 epochs with various batch sizes ranging from 32 to 1024. As shown in Table. 4, even if the batch size is as low as 32 or 64, INTL still maintains good performance. At the same time, when the batch size increases, the accuracy of INTL is also improved. These results indicate that INTL has good robustness to batch sizes and can adapt to various scenarios that constrain the training batch size.

Table 4: Effect of batch sizes for INTL. We train 100 epoch on ImageNet and provide the Top-1 accuracy using linear evaluation. The embedding dimension is fixed to 8192.

| Bs | 32 | 64 | 128 | 256 | 512 | 1024 |
|---|---|---|---|---|---|---|
| acc.(%) | 64.2 | 66.4 | 68.1 | 68.7 | 69.5 | 69.7 |

**Embedding dimension.** Embedding dimension, the output dimension of the projection, is also a key element for most self-supervised learning methods, which may have a significant impact on training results. As illustrated in [44], Barlow Twins is very sensitive to embedding dimension and it requires a large dimension (e.g. 8192 or 16384) to work well. We also test the robustness of INTL to embedding dimensions. Following the setup of [7] and [44], we train INTL on ImageNet for 300 epochs with the dimension ranging from 64 to 16384. As shown in Figure. 4, even when the embedding dimension is low as 64 or 128, INTL still achieves good results. These results show that INTL also has strong robustness to embedding dimensions.

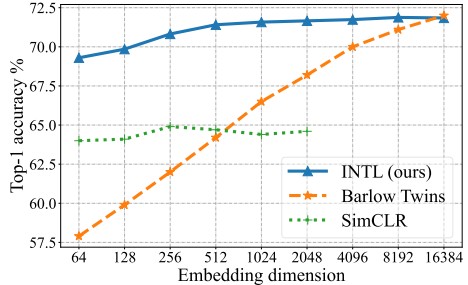

Figure 4: Ablation experiments for varying embedding dimensions. The batch size is fixed to 256.

## 6 Conclusion and Limitation

In this paper, we proposed spectral transformation (ST) framework to modulate the spectrum of embedding and to seek for functions beyond whitening that can avoid dimensional collapse. Our proposed IterNorm with trace loss (INTL) is well-motivated, theoretically demonstrated, and empirically validated in avoiding dimension collapse. Comprehensive experiments have shown the merits of INTL for achieving state-of-the-art performance for SSL in practice. We showed that INTL provides the equal-eigenvalues constraint on the covariance matrix of *embedding*, which is a stronger constraint than hard whitening (the full-rank constraint), but a weaker constraint than soft whitening (the whitening constraint). This preliminary but new results provides a potential way to understand and compare SSL methods.

**Limitation.** Our work only explores the mechanism of ST using power function and Newton's iteration for SSL. As a general concept, we believe that more functions in our ST framework can be designed to avoid collapse in the future. Besides, our theoretical work mainly revolves around the ST using Newton's iteration, without providing theoretical analysis for more general ST. There are still mysteries about modulation of the spectrum in more general ST during backpropagation.

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
