# OpenReview forum: "Modulate Your Spectrum in Self-Supervised Learning"
_NeurIPS.cc/2023/Conference — Submitted to NeurIPS 2023_

### Official Review · Reviewer_Fnki · 2023-07-05

**Soundness:** 2 fair
**Presentation:** 3 good
**Contribution:** 2 fair
**Rating:** 5
**Confidence:** 5

**Summary:**

Feature collapse is a major problem in contrastive-base self-supervised learning. To address this issue, the paper introduces the spectral transformation framework, which aims to mitigate the aforementioned problem. Generally speaking, the spectral transformation expands upon the extensively employed whitening transformation discussed in the literature and treats it as a special case. Furthermore, the paper proposes the IterNorm approach featuring trace loss to enhance the performance of self-supervised models. The experimental results validate the effectiveness of the IterNorm with trace loss (INTL) on various widely-used SSL-evaluation benchmarks, including but not limited to linear evaluation, classification, and object detection.

**Strengths:**

1)	The paper addresses a fundamental challenge in the context of contrastive-based self-supervised learning, and proposes an elegant method with theoretical guarantees. Specifically, the proposed framework extends the widely used whitening operation discussed in prior literature, treating it as a special instance, thereby enhancing the theoretical contribution of this work.

2)	 The experimental results are promising. The proposed method is evaluated on several SSL-evaluation benchmarks, including ImageNet linear classification (achieving 76.6% accuracy on ResNet-50), COCO object detection (achieving a score of 41.2 on ResNet50-C4), and significantly surpasses the previous state-of-the-art methods.

3)	The presentation of the paper is good, with a clear elaboration of the motivation and contribution of the proposed method.


**Weaknesses:**

1)	The comparison between the proposed method and baseline is unfair. Firstly, the proposed framework employs a 3-layer projection head with a substantially larger dimension (8192), which significantly increases both computation and memory costs. This discrepancy makes it hard to conduct a fair comparison between the proposed method and the baseline approach.

2)	As mentioned earlier, the proposed method incurs nearly twice the computation time per epoch and peak memory per GPU compared to the naive contrastive baseline method MoCo/MoCo-v2.

3)	It is worth noting that the multi-crop strategy employed in the paper is highly similar to that used in [a], including the crop-nums and crop-sizes. It is unclear why this was not cited in the paper.

4)	It should be noted that the multi-crop strategy employed in [a] can also enhance the detection and classification results. Therefore, the absence of this information in the comparison makes it difficult to conduct a fair evaluation.

[a] Wang, X., & Qi, G. J. (2022). Contrastive learning with stronger augmentations. IEEE transactions on pattern analysis and machine intelligence, 45(5), 5549-5560.


**Questions:**

1)	The experiments conducted in the paper are based solely on the ResNet-50 network architecture, it would be beneficial if the authors could provide results based on transformer backbones, given their widespread usage in the self-supervised learning domain.

2)	As a theoretical-style paper, it is important to have a fair and direct comparison between the proposed method and the baseline approach. However, the utilization of several tricks and unfair experimental settings in the paper makes it challenging to determine the actual degree of improvement gained from the original technical contribution. Furthermore, such experimental settings make it hard to follow the paper's methodology.

**Limitations:**

yes

---

> ### Author Rebuttal · Authors · 2023-08-08
>
> ## Response to Reviewer Fnki
>
> We thank the reviewer for the encouraging and insightful comments. Please find our responses to specific concerns and questions below.
>
> **Concern 1:** The comparison between the proposed method and baseline is unfair. ... the baseline approach.
>
> **Response:** Thanks for your comments. The reason why we set the projection dimension as 8192 is that our initial experiments followed the settings of VICReg and Barlow Twins, both of which use a dimension of 8192 for the projection. Compared to a projection dimension of 2048, using a projection dimension of 8192 can bring about a 0.14% improvement in top-1 accuracy for INTL. Therefore, we followed this setting in subsequent experiments on ImageNet. We report that using a projection dimension of 8192 requires approximately 18% additional GPU memory and 2% time per epoch compared to using the one of 2048.
>
> We think it is difficult to apply identical hyperparameters for all baselines and our method. Because each self-supervised method has its own characteristics, it may be unfair to use the same hyperparameters for all methods. For example, a small projection dimension of 512 or 1024 is suitable for W-MSE and BYOL but will have degenerated performance for VICReg and Barlow Twins. Meanwhile, a large projection dimension of 8192 fails to train W-MSE because W-MSE requires the batch size to be larger than the projection dimension. We note that solo-learn [11], which is designed as a framework for integrating and reproducing various self-supervised methods, also adopts different hyperparameters for different methods.
>
> **Concern 2:** As mentioned earlier, ... MoCo/MoCo-v2.
>
> **Response:** In our experiments, our INTL can also uses Exponential Moving Average (EMA), like the Moco-v2. Our INTL with EMA requires a total of around 23.6 GB GPU memory and 24min46s running time per epoch (Table G of supplementary materials),  which is comparable to 20 GB memory and 23min11s running time per epoch required by MoCo-v2 that also uses EMA. Although requiring a bit more memory and computation time, our method achieves a 2.1% improvement in top-1 accuracy on ImageNet (74.3% v.s. 72.2%) compared to MoCo-v2, as shown in Table D of our supplementary materials. Meanwhile, compared to other methods that use an additional predictor (such as BYOL), or use eigen-decomposition (such as W-MSE), our proposed INTL not only reduces memory and computation time but also achieves performance improvement.
>
> **Concern 3:** It is worth noting that the multi-crop strategy ... why this was not cited in the paper.
>
> **Response:** Thanks sincerely for your constructive suggestions. We mainly draw inspiration from SwAV to set our multi-crop strategy, but did not realize that the similar variant had been proposed in paper [a]. We will cite [a] in the revised version.
>
> **Concern 4:** It should be noted that ... makes it difficult to conduct a fair evaluation.
>
> **Response:** Note that the results of all methods on CIFAR-10/100 and ImageNet-100, shown in Table 2 of our submission, are trained without multi-crop strategy, following the settings of solo-learn [11]. We also provide the classification results on ImageNet without multi-crop strategy in Table D of our supplementary materials. Without multi-crop, INTL can achieve a top-1 accuracy of 74.3% on ImageNet. All these results indicate that INTL can also achieve better performance or is on par with other SOTA methods without multi-crop strategy. Note that Multi-crop is a common strategy used by SSL methods (such as SwAV, DINO, W-MSE, and so on) to boost performance, so we also apply it to INTL.
>
> Following your insightful comments, we conduct additional experiments to evaluate the detection results of INTL without multi-crop. We train INTL w\ and w\o multi-crop on ImageNet for 200 epochs and then transfer pre-trained backbones to other tasks. The results are shown in the table below. It shows that multi-crop strategy can slightly enhance our detection results, while our method without multi-crop can still perform better than SwAV with multi-crop.
>
> |        Method         |                        COCO detection                        |                      COCO segmentation                       |
> | :-------------------: | :----------------------------------------------------------: | :----------------------------------------------------------: |
> |                       | $AP_{50}$ &emsp;&emsp;&emsp; $AP$ &emsp;&emsp;&emsp; $AP_{75}$ | $AP_{50}$ &emsp;&emsp;&emsp;$AP$ &emsp;&emsp;&emsp; $AP_{75}$ |
> | SwAV (w\ multi-crop) |  60.2 &emsp;&emsp;&emsp;39.8 &emsp;&emsp;&emsp;43.0  |  56.6 &emsp;&emsp;&emsp; 34.6 &emsp;&emsp;&emsp; 36.8 |
> | INTL (w\o multi-crop) |  $60.9_{±0.08}$ &nbsp; $40.7_{±0.09}$ &nbsp; $43.7_{±0.17}$  |  $57.3_{±0.08}$ &nbsp; $35.4_{±0.05}$ &nbsp; $37.6_{±0.14}$  |
> | INTL (w\ multi-crop)  |  $61.2_{±0.08}$ &nbsp; $41.2_{±0.12}$ &nbsp; $44.7_{±0.19}$  |  $57.8_{±0.04}$ &nbsp; $35.7_{±0.04}$ &nbsp; $38.1_{±0.12}$  |
>
> **Question 1:** The experiments ... on transformer backbones, given their widespread usage in the self-supervised learning domain.
>
> **Response:**  Thanks for your suggestion, and we conduct additional experiments using ViT backbones. INTL can also perform better than other baselines. Please check Figure 1 of the rebuttal pdf for details.
>
> **Question 2:** As a theoretical-style paper, ... Furthermore, such experimental settings make it hard to follow the paper's methodology.
>
> **Response:** We sincerely thank the reviewer for the insightful suggestions. Our main theoretical claim is that INTL can avoid dimensional collapse and modulate the spectrum of embedding towards an equal-eigenvalue distribution. This claim is empirically validated in Section 4.2. In terms of the empirical comparison on standard SSL benchmark, we believe our experimental setup is fair, and please see the responses of Concern 1 and Concern 4.

---

### Official Review · Reviewer_SEtq · 2023-07-06

**Soundness:** 3 good
**Presentation:** 2 fair
**Contribution:** 3 good
**Rating:** 5
**Confidence:** 4

**Summary:**

For self-supervised learning, this paper proposes a spectral transformation (ST) framework to modulate embedding, seeking functions other than whitening that can avoid dimensional collapse. The authors propose IterNorm with trace loss and provide a lot of theoretical and experimental analysis. The results show its effectiveness and advancement.

**Strengths:**

1. This paper provides intuitive experimental verification, extending the whitening transformation to a more general spectral transformation.
2. For the proposed IterNorm with trace loss, this paper conducts rigorous theoretical analysis and practice, showing that it is effective for avoiding collapse.
3. The proposed method can achieve good results without relying on large batches, which can be comparable to supervised performance under regular settings.


**Weaknesses:**

1.	The motivation for the proposed IterNorm with trace loss is unclear. The authors empirically observe that IterNorm suffers from severe dimensionality collapse, but do not state the limitations of existing SSL methods in this case.
2.	For the ablation study of batch size, Table 4 shows that the performance of the proposed method increases significantly as the batch size increases, indicating that the method is also sensitive to the batch size. In addition, SimCLR, SwAV and other methods require a batch size of 4096. Table 4 should provide the corresponding results with a batch size of 4096 and the sensitivity of other methods to the batch size.
3.	Among the existing SSL methods compared in the experimental part, few new methods since 2022 are covered. More state-of-the-art method comparisons need to be added.


**Questions:**

Please see above weaknesses.

**Limitations:**

The authors have explained some of the limitations in the paper, and others can be seen in weaknesses.

---

> ### Author Rebuttal · Authors · 2023-08-08
>
> ## Response to Reviewer SEtq
>
> We thank the reviewer for the encouraging and insightful comments. Please find our responses to specific concerns below.
>
> **Concern 1:** The motivation for the proposed IterNorm with trace loss is unclear. The authors empirically observe that IterNorm suffers from severe dimensionality collapse, but do not state the limitations of existing SSL methods in this case.
>
> **Response:** The main motivation of this paper is to address the dimensional collapse problem in SSL. We generalize the whitening transformation to spectral transformation (ST), which could be a general framework to design algorithms to avoid dimensional collapse in SSL, and we indeed design INTL (IterNorm + trace loss) as a new instance of ST that avoid dimensional collapse theoretically.
> Note that the motivation to use IterNorm is to avoid the numerical instability of ‘ST using power functions’ (including existing whitening-based methods), which is shown in Lines 201-210 of our submission.  The motivation to use INTL is that IterNorm-only also suffers dimensional collapse unexpectedly in SSL, but adding trace loss can avoid it based on our theoretical analyses and empirical validation.
>
> **Concern 2:** For the ablation study of batch size, Table 4 shows that the performance of the proposed method increases significantly as the batch size increases, indicating that the method is also sensitive to the batch size. In addition, SimCLR, SwAV and other methods require a batch size of 4096. Table 4 should provide the corresponding results with a batch size of 4096 and the sensitivity of other methods to the batch size.
>
> **Response:** Following your comments, we conduct experiments to compare the performances of  SimCLR, SwAV,  and INTL pre-trained on ImageNet for 100 epochs with respect  to the batch sizes. The results are shown in the table below. Due to training with a batch size of 4096 requiring approximately 512GB GPU memory, we are unable to train the corresponding models with our workstation of 4 A100-PCIE-40GB GPUs. The results of other methods using a batch size of 4096 are from the corresponding papers.
> One important phenomenon in identifying sensitivity to batch sizes is that when the batch size decreases from 128 to 64, the top-1 accuracy of SimCLR and SwAV significantly degraded by nearly 10%, while the top-1 accuracy of our INTL slightly  degraded by 1.7%, which performs much better than the other two methods. These experimental results indicate that INTL has good robustness to batch sizes. We will include these results in the revised version.
>
> | Method |  32  |    64    |   128    |   256    |   512    |   1024   | 4096 |
> | :----- | :--: | :------: | :------: | :------: | :------: | :------: | :--: |
> | SimCLR |  -   |   52.9   |   62.3   |   63.0   |   65.1   |   66.0   | 66.5 |
> | SwAV   |  -   |   53.6   |   63.7   |   65.9   |   66.3   |   66.3   | 66.5 |
> | INTL   | 64.2 | **66.4** | **68.1** | **68.7** | **69.5** | **69.7** |  -   |
>
> **Concern3:** Among the existing SSL methods compared in the experimental part, few new methods since 2022 are covered. More state-of-the-art method comparisons need to be added.
>
> **Response:** Sincerely thanks for your constructive suggestions. We conduct additional experiments to compare the state-of-the-art methods, including Zero-CL (ICLR 2022) and CW-RGP (NeurIPS 2022), following the same setups as in Table 2.  The results are shown in the table below.  We find that INTL also outperforms these methods. We would like to add the results in the revised version.
>
> |    Method    |              CIFAR-10               |              CIFAR-100               |            ImageNet-100             |
> | :----------: | :---------------------------------: | :----------------------------------: | :---------------------------------: |
> |              |   top-1 &nbsp; 5-nn &nbsp; top-5    |    top-1 &nbsp; 5-nn &nbsp; top-5    |   top-1 &nbsp; 5-nn &nbsp; top-5    |
> |   Zero-CL (ICLR 2022)    |   90.81 &nbsp; 87.51 &nbsp; 99.77   |   70.33 &nbsp; 59.21 &nbsp; 92.05    |  79.26  &nbsp; 71.18 &nbsp; 94.98   |
> |    CW-RGP (NeurIPS 2022)  |   92.03 &nbsp; 89.67 &nbsp; 99.73   |   67.78 &nbsp; 58.24  &nbsp; 90.65   |   76.96 &nbsp;68.46 &nbsp; 93.76    |
> | INTL (ours)  | **92.60 &nbsp; 90.03 &nbsp; 99.80** | **70.88 &nbsp; 61.90 &nbsp; 92.13** | **81.68 &nbsp; 73.46 &nbsp; 95.42** |

---

### Official Review · Reviewer_tQK6 · 2023-07-06

**Soundness:** 3 good
**Presentation:** 2 fair
**Contribution:** 4 excellent
**Rating:** 5
**Confidence:** 3

**Summary:**

This submission proposes a spectral transformation for redundancy-reduction-based (a.k.a. whitening-based) self-supervised learning (SSL).
Spectrum-domain modulation is helpful in preventing collapsing caused by representations' rank deficiency.
Since the whitening operation can be seen as a square-root transformation on spectrum, it can be generalized to other
power-function transformation.
To avoid numerical instability of matrix decomposition in computing spectra, an approximation based on Newton’s iteration method IterNorm
is proposed. While IterNorm by itself suffers from dimensional collapse, its combination with trace loss, the term to encourage the spectrum to have equal-eigenvalue distribution, works well to avoid collapse.
Experiments on ImageNet, CIFAR-10/100, and COCO show that the proposed method outperforms existing state-of-the-art methods.

**Strengths:**

- The idea of spectrum-domain modulation seems intuitive to avoid dimensional collapse caused by rank deficiency of representations.
- State-of-the-art SSL performances using ResNet50 were achieved.
- Especially excellent performance avoiding collapse in small batch sizes is practical for low-resource training.

**Weaknesses:**

- The difference between existing whitening-based methods and power-function modulation with p = 0.5: Sec 3.2 empirically shows that p == 0.5 (== 1/2) is a good value. At the same time, p4 l 170 says, "whitening is a special instance of spectral transformation, where g(·) is a power function g(λ) = λ^{− 1/2}". This confused me with what difference this generalization made from usual whitening.

- Trace loss + BarlowTwins/VICReg?: It is said that IterNorm works well only when combined with trace loss.
This naturally implies that the source of success is trace loss itself rather than IterNorm. Is it possible to combine trace loss with existing whitening-based methods part from IterNorm?



**Questions:**

- p4 l151 "a function over embedding Z during forward pass, and modulates the spectrum of embedding Z implicitly during backward pass when minimizing MSE loss": I could not follow this part; what kind of modulation is done in the backward path? More concrete explanation is preferable.

**Limitations:**

Limitations or social impact are not discussed, but I do not have particular concerns.

---

> ### Author Rebuttal · Authors · 2023-08-08
>
> ## Response to Reviewer tQK6
>
> We thank the reviewer for the encouraging and insightful comments. Please find our responses to specific concerns and questions below.
>
> **Concern 1:** The difference between existing whitening-based methods and power-function modulation with $p = 0.5$: Sec 3.2 empirically shows that $p==0.5 (== 1/2)$ is a good value. At the same time, p4 l 170 says, 'whitening is a special instance of spectral transformation, where g(·) is a power function $g(λ) = λ^{− 1/2}$'. This confused me with what difference this generalization made from usual whitening.
>
> **Response:** In our Spectral Transformation framework,  $g(λ)$ can be any function which finnally maps the eigenvalue $\lambda$ to  $\lambda g^2(\lambda)$. Therefore, both whitening ($g(\lambda)=\lambda^{-1/2}$) and power function ($g(\lambda)=\lambda^{-p}, p \in \mathbb R$) are special cases of ST.  Note that the mapping function $g(λ)$ of IterNorm with $T$ iteration number is $f_T(\frac{\lambda}{tr(\Sigma)}) / \sqrt{ tr(\Sigma)}$ (Line 218), which is not a power function, but an ST by definition (defined over the iterative function $f_{k+1}(x) = \frac{3}{2} f_{k}(x)-\frac{1}{2} x {f_{k}}^3(x), k \geq 0$;  $f_{0}(x) = 1.$ shown in Line 216); Besides, different $T$ of IterNorm implies different mapping function $g(λ)$ of IterNorm. Note that we want to show that there exist other functions for ST except whitening that can also work to avoid collapse in Section 3.2 (e.g., the power function for ST with $p = 0.45$ or $0.55$ in Figure 2). Perhaps our unclear description caused your misunderstanding, and we will further refine this section.
>
> **Concern 2:** Trace loss + BarlowTwins/VICReg?: It is said that IterNorm works well only when combined with trace loss. This naturally implies that the source of success is trace loss itself rather than IterNorm. Is it possible to combine trace loss with existing whitening-based methods part from IterNorm?
>
> **Response:** Following your insightful comments, we conduct experiments on CIFAR-10/100 and ImageNet-100 to evaluate the performance of adding trace loss to BarlowTwins/VICReg,  following the same setup in Table 2 of our paper (except that we train models on CIFAR-10/100 for 200 epochs and on ImageNet-100 for 100 epochs due to time limit of the rebuttal period). We set the coefficient of trace loss to 0.01, which is empirically applicable for both methods. The results are shown in the table below. We find that adding trace loss to Barlow Twins is feasible and can slightly improve the performance, but adding it to VICReg heavily reduces the performance, especially on ImageNet-100. We conjecture the reason is likely that adding trace loss to these two methods will affect the intensity of regularization, which may disrupt the original balance and lead to a decrease in performance, or achieve a better balance and boost performance.
>
>
> |          Method           |                CIFAR-10                 |              CIFAR-100               |            ImageNet-100             |
> | :-----------------------: | :-------------------------------------: | :----------------------------------: | :---------------------------------: |
> |                           |     top-1 &nbsp; 5-nn &nbsp; top-5      |    top-1 &nbsp; 5-nn &nbsp; top-5    |   top-1 &nbsp; 5-nn &nbsp; top-5    |
> |       Barlow Twins        |   80.43 &nbsp; **76.68** &nbsp; 99.05   |   51.60 &nbsp; 42.71 &nbsp; 80.37    |   58.34 &nbsp; 50.21&nbsp; 83.46    |
> | Barlow Twins + trace loss | **80.45** &nbsp; 76.32 &nbsp; **99.15** | **51.66 &nbsp; 43.94  &nbsp; 81.37** | **59.78 &nbsp; 50.45 &nbsp; 83.80** |
> |                           |                                         |                                      |                                     |
> |          VICReg           |   **83.14 &nbsp; 79.62 &nbsp; 99.23**   | **55.96 &nbsp; 46.71 &nbsp;  83.37** | **66.01 &nbsp;57.76 &nbsp; 89.14**  |
> |    VICReg + trace loss    |     81.67 &nbsp; 78.74 &nbsp; 99.06     |   54.75 &nbsp; 46.24 &nbsp;  82.98   |   63.54 &nbsp; 55.18 &nbsp; 86.34   |
>
>
> **Question 1:** ‘a function over embedding Z during forward pass, and modulates the spectrum of embedding Z implicitly during backward pass when minimizing MSE loss’: I could not follow this part; what kind of modulation is done in the backward path? More concrete explanation is preferable.
>
> **Response:** Thanks for your suggestion. We do not address the exact formulation of how the spectrum of embedding is modulated in each optimization step (backward path), but we address the optimization direction of the spectrum of embedding when using gradient-based methods to modulate the spectrum. That is, what spectrum of embedding will be modulated to be during the course of training.
> For example, whitening loss encourages the embedding to be full-rank [40] during the course of training. That is, the spectrum of embedding will be modulated to be full-rank when we update the embedding using gradient-based optimization methods. In this paper, we theoretically prove that INTL promotes the equality of all eigenvalues of the covariance matrix of embedding $\mathbf{Z}$ during the course of optimization. This kind of equal-eigenvalue spectrum of embedding is what INTL wants to modulate to be during the backward pass. Following your insightful comments, we will provide a more concrete explanation of the modulation in the revised version.

---

> > ### Comment · Reviewer_tQK6 · 2023-08-19
> >
> > I appreciate the Authors' detailed response.
> > The response well answered my concerns, including Concern 1 which was based on my misunderstanding.
> > I recommend including full-iteration comparisons of INTL vs Barlow Twins + trace loss & VICReg + trace loss in the camera-ready.
> > I now safely vote for acceptance and keep my initial rating.

---

> > > ### Author Response · Authors · 2023-08-19
> > > **Thank you for the response**
> > >
> > > Dear reviewer tQK6,
> > > We sincerely appreciate you taking the time to review our responses and vote for acceptance of our paper. We will carefully follow all of your comments and include them in the revised version to improve this paper.
> > >
> > > Best wishes to you.
> > >
> > > Authors of Paper 6124

---

### Official Review · Reviewer_Ckne · 2023-07-07

**Soundness:** 3 good
**Presentation:** 3 good
**Contribution:** 2 fair
**Rating:** 3
**Confidence:** 3

**Summary:**

The paper studies the dimension collapse problem in image self-supervised learning. Previously, people use image space data augmentation/momentum/stop gradient and so on to address this problem. This paper proposes to address the problem via spectrum transformation (ST) in the feature space instead (aka balancing the eigenvalue of covariance matrix while keeping eigenvectors the same in mini-batches).



**Strengths:**

1.	Its ST generalized previous whitening approach (aka make the covariance matrix to be identity), and extends it to power functions (aka p \in R instead of p = 0.5).
2.	For the calculation of whitening matrix, the paper proposes a less expensive approach to calculate (IterNorm). And the paper analyzes how the singular values will change using IterNorm. But to really make things work, the paper adds a trace loss (INTL).
3.	The paper shows INTL reaches SOTA performance on image classification and object detection tasks. And INTL is less sensitive to batch size.

**Weaknesses:**

1. The **major concern** for this paper is that, it resembles quite a few similarities in the key ideas to a recent AAAI work: "Spectral Feature Augmentation for Graph Contrastive Learning and Beyond" (https://arxiv.org/abs/2212.01026), but the AAAI work has not been cited or discussed in this work, making the contribution of this work a bit incremental:

a. The motivations are similar: deal with feature collapse/dimension collapse via change feature space’s spectrum.

b. Similar pipeline, the AAAI paper apply SFA after encoder and before projection, this work applies ST in the end of the pipeline. the AAAI work injects noise for better augmentation while the paper does not. This paper finally needs to add an additional trace loss to make ST work, which may play a similar role to the injected noise.

c. Iterative normalization to calculate \Sigma_{-0.5} is like appendix C8 algorithm 2. But with a slightly different formulation. The AAAI paper calculates \Sigma_{-0.5} and \Sigma_{0.5} at the same time. This work only calculates \Sigma_{0.5} with a slightly different iteration function.

d. This work’s power functions for ST are like AAAI’s MaxExp(F) and Matrix Square Root, but this work’s approach does not inject a Gaussian noise.

e.Both experiments on SSL on ImageNet. The AAAI work explored both graphs and images. This work focuses more on images, so it also explores CIFAR, and also object detection tasks. But AAAI work deals with contrastive SSL (InfoNCE) while this work uses a non-contrastive approach (normalized MSE loss).

2. It states to explore other possible spectrum transformation approaches but only extends from whitening (p=0.5) to power (p \in R) with a limited range of working p around 0.5. This does not validate its statement to generalize ST.

3. Directly using ST (IterNorm) does not work, and finally needs to add trace norm (INTL). It is not shown that if only using the implicit trace norm without explicit ST, whether the performance will still be good. This does not validate its need to do explicit ST.


**Questions:**

The notation in Figure 1 is not clear: are X, Z matrices? The notation of Z in Figure 1 and Equation are inconsistent. It would also be better to add notation section since there are many similar symbols.

---

> ### Author Rebuttal · Authors · 2023-08-08
>
> ## Response to Reviewer Ckne
>
> We thank the reviewer for the encouraging and insightful comments. Please find our responses to specific concerns and questions below.
>
> **Concern 1:** The major concern for this paper is that, it resembles quite a few similarities in the key ideas to a recent AAAI work.
>
> **Response:** We thank the reviewer for pointing out this recently related work [AAAI 2023] and describing the similarities and differences between our work and the [AAAI 2023]. Here, we further complement the differences based on the view of the reviewer:
>
> 1. In terms of point (b) of the Reviewer, we address that IterNorm (an instance of ST) requires an additional trace loss to work, but whitening (also an instance of ST) can work well without an additional trace loss;
> 2. In terms of point (d) of the Reviewer, we address that we use the power function in Section 3.2 as an example to illustrate that there exists ST beyond whitening can avoid dimensional collapse. Note that $g(\lambda)$ of IterNorm with $T$ iteration number is $f_T(\frac{\lambda}{tr(\Sigma)}) / \sqrt{ tr(\Sigma)}$ (Line 218, and $f_T(\cdot) $is defined in Line 216), which is not a power function, but an ST by definition;
> 3. We theoretically prove that IterNorm with trace loss (INTL) can avoid collapse and modulate the spectrum of embedding towards an equal-eigenvalue distribution.
>
> We would like to cite the related work [AAAI 2023] and take credit for its contribution to SSL using spectral analyses in our revised version.
>
> **Concern 2:** It states to explore other possible spectrum transformation approaches but only extends from whitening (p=0.5) to power (p \in R) with a limited range of working p around 0.5. This does not validate its statement to generalize ST.
>
> **Response:** We highlight that the power function is just a kind of ST, and we use the power function in Section 3.2 as an example to illustrate that there exists ST beyond whitening that can avoid dimensional collapse. Note that the mapping function $g(\lambda)$ of IterNorm with $T$ iteration number is $f_T(\frac{\lambda}{tr(\Sigma)}) / \sqrt{ tr(\Sigma)}$ (Line 218), which is not a power function, but an ST by definition (defined over the iterative function $f_{k+1}(x) = \frac{3}{2} f_{k}(x)-\frac{1}{2} x {f_{k}}^3(x), k \geq 0$;  $f_{0}(x) = 1.$ shown in Line 216); Besides, different $T$ of IterNorm implies different mapping function  of IterNorm.
>
> **Concern 3:** Directly using ST (IterNorm) does not work, and finally needs to add trace norm (INTL). It is not shown that if only using the implicit trace norm without explicit ST, whether the performance will still be good. This does not validate its need to do explicit ST.
>
> **Response:** Following your insightful comments, we conduct additional experiments on CIFAR-10 to validate the performance of using trace loss without IterNorm. We use the same experimental setup in Table 2 of our paper, except that the models are only pre-traind for 200 epochs on CIFAR-10. The results are shown in the table below.
>  We observe that trace loss without IterNorm fails to avoid dimensional collapse. Note that IterNorm ensures the covariance matrix of transformed output $\widehat{\mathbf{Z}}$ having eigenvalues between $(0, 1)$. Trace loss combining IterNorm encourages all the eigenvalues of the covariance matrix of $\widehat{\mathbf{Z}}$ towards $1$ and further promotes the equality of all eigenvalues of the covariance matrix of embedding $\mathbf{Z}$ during the course of optimization, as shown by Theorem 2. This provides a theoretical guarantee to avoid dimensional collapse.  We will add these results to validate the necessity of ST in Section 4 of our paper.
>
> | Method         |             w\ IterNorm             |          w\o IterNorm           |
> | :------------- | :---------------------------------: | :-----------------------------: |
> |                |    top-1 &nbsp; 5-nn &nbsp;top-5    |  top-1 &nbsp; 5-nn &nbsp;top-5  |
> | w\ trace loss  | **90.75 &nbsp; 87.58 &nbsp; 99.71** | 16.15 &nbsp; 12.34 &nbsp; 58.65 |
> | w\o trace loss |   36.15 &nbsp; 31.34 &nbsp; 73.65   | 10.00 &nbsp; 10.00 &nbsp; 50.00 |
>
> **Question 1:** The notation in Figure 1: are X, Z matrices?
>
> **Response:** Yes, $\mathbf{X}$ and $\mathbf{Z}$ are matrices. In Lines 142~144, we illustrate that $\mathbf{Z}$ is the corresponding mini-batch embedding, given mini-batch input $X$. We use the capital letter to represent the mini-batch (a form of matrix) of input ($\mathbf{X}$), embedding ($\mathbf{Z}$), and transformed output ($\widehat{\mathbf{Z}}$).

---

> ### Author Response · Authors · 2023-08-18
> **We thank the reviewer again for the valuable feedback and happy to address any remaining concerns**
>
> We extend our sincere gratitude to the reviewer for their valuable time and insightful feedback. We value the constructive feedback and hope that our responses have appropriately addressed all the concerns. We really appreciate the valuable time to respond to our feedbacks based on the reviewer's comments. Further, we are happy to address any remaining concerns.

---

### Official Review · Reviewer_QdWo · 2023-07-10

**Soundness:** 3 good
**Presentation:** 4 excellent
**Contribution:** 3 good
**Rating:** 6
**Confidence:** 2

**Summary:**

This paper addresses the dimensional collapse problem in self-supervised learning and proposes a framework to moderate the spectrum of embedding. Besides, a new spectral transformation variant, called IterNorm with trace loss (INTL), is proposed that can avoid collapse and modulate the spectrum of embedding towards an equal-eigenvalue distribution. Experiments on various tasks verify the effectiveness of the proposed method.

**Strengths:**

- The proposed method looks technically sound.

- Extensive experiments have been conducted to verify the effectiveness by comparing with the existing methods in the field.

- The presentation is great.

**Weaknesses:**

This looks like a good paper. I do not have much to complain about (possibly due to I do not work in this direction). I just have two minor questions.

- How do you balance the MSE loss and IterNorm trace loss? Are there coefficients in the two loss terms? If there are, how do you set the coefficient values? Do they keep fixed across all the datasets?

- The proposed method INTL is based on the empirical finding that when p is in the neighborhood of 0.5, e.g. 0.45 ∼ 0.55, the model can avoid collapse. Is there any intuitive explanation or theoretic analysis for this phenomenon?

**Questions:**

See the weakness above.

**Limitations:**

See the weakness above.

---

> ### Author Rebuttal · Authors · 2023-08-08
>
> ## Response to Reviewer QdWo
>
> We thank the reviewer for the encouraging and insightful comments. Please find our responses to specific questions below.
>
> **Question 1:** How do you balance the MSE loss and IterNorm trace loss? Are there coefficients in the two loss terms? If there are, how do you set the coefficient values? Do they keep fixed across all the datasets?
>
> **Response:** Yes, there is a coefficient between the MSE loss and IterNorm trace loss. We set the loss function of INTL as $\mathcal{L}\_{INTL}=\mathcal{L}\_{MSE}+\beta \cdot \mathcal{L}\_{trace}$. In our experiments, we observe that the coefficient $\beta$ that obtains good performance has relevant to the batch size. So we empirically regress $\beta$ with various batch sizes and obtain that $\beta=0.01*(log_{2} bs-3)$ where $bs$ means the batch size and $bs>8$. We keep the coefficient $\beta$ fixed in this form (i.e.,  $\beta$ **is determined given the batch size**) across all the datasets and architectures, so our INTL can be directly applied to other datasets and models without tuning the coefficient. These settings and descriptions can also be found in Section C ‘Algorithm of INTL’ of our supplementary materials.
>
> **Question 2:** The proposed method INTL is based on the empirical finding that when $p$ is in the neighborhood of $0.5$, e.g. $0.45 ∼ 0.55$, the model can avoid collapse. Is there any intuitive explanation or theoretic analysis for this phenomenon?
>
> **Response:** We note that a recent work [40] implied that whitening loss can be decomposed into two asymmetric losses $\mathcal{L} = \frac{1}{m} \| \phi(\mathbf{Z}\_{1})\mathbf{Z}\_{1} - (\widehat{\mathbf{Z}}\_{2})\_{st}  \|\_{F}^2 + \frac{1}{m} \| \phi(\mathbf{Z}\_{2})\mathbf{Z}\_{2} - (\widehat{\mathbf{Z}}\_{1})_{st}  \|\_{F}^2$, where $\phi(\mathbf{Z})$ is the whitening matrix of $\mathbf{Z}$, and $\widehat{\mathbf{Z}}$ is the whitening output. Each asymmetric loss can be viewed as an online network to match a whitened target $\widehat{\mathbf{Z}}$. As a general form of whitening, our ST can also apply this decomposition to the loss function. Our intuitive explanation is as follows:
>
> When $p$ is in the neighborhood of $0.5$, e.g. $0.45 ∼ 0.55$, $\widehat{\mathbf{Z}}$ has a well-conditioned spectrum that each eigenvalue approaches 1. In this case, $\widehat{\mathbf{Z}}$ is a good target for $\phi(\mathbf{Z})\mathbf{Z}$ to match so that the embedding $\mathbf{Z}$ can learn a good spectrum to avoid collapse.  For example, if $\mathbf{Z}$ suffers dimensional collapse,  $\phi(\mathbf{Z})\mathbf{Z}$ will be low-rank and is impossible to exactly match the full-rank $\widehat{\mathbf{Z}}$.
>
> On the contrary, when $p$ deviates from $0.5$ to a certain extent, the spectrum of the transformed output $\widehat{\mathbf{Z}}$ is not well-conditioned and is not a good target in representation. We believe that more theoretical work in this part is worth exploring in the future.

---

### Official Review · Reviewer_TM8b · 2023-07-12

**Soundness:** 3 good
**Presentation:** 4 excellent
**Contribution:** 2 fair
**Rating:** 6
**Confidence:** 3

**Summary:**

This paper tackles the dimension collapse problem in self-supervised learning. The authors propose spectral transformation which can be served as an alternative for the whitening function in avoiding dimensional collapse. Further, they propose a new instance of ST, called IterNorm with trace loss (INTL), and prove that INTL can avoid collapse. Results on several SSL benchmarks show promising improvements over previous methods.

**Strengths:**

- The paper is well-organized with a clear story and justification.
- The paper aims at analyzing the dimension collapse problem and prove the previous whitening function is a special case of spectrum transformation (ST).
- The authors claim that the numerical instability can be solved if a spectral transformation can modulate the spectrum without explicitly calculating λ or U. Then, they propose Iterative Normalization with Trace Loss as a solution.
-  The performance on Imagenet, CIFAR and COCo shows improvements over previous methods such as Btwins, and VICreg.


**Weaknesses:**

1. Comparing methods in the Experiment section:
As the proposed method is claimed to be the general form of whitening function in SSL, I am wondering why the authors did not put the performance comparison against whitening loss [20, 40]. As [20, 40] have demonstrated the performance gains over Btwins, it is hard to tell how the INTL it better than [20, 40].

2. Concerns about novelty.
In my view, section 3.2 Spectral Transformation proves the effectiveness of Whitening. The general form of ST is $g(\lambda) = \lambda^{-p}$ and the authors claim that
> it seems to perform well to avoid collapse although the transformed output is not ideally whitened when p is in the neighborhood of 0.5, e.g. 0.45 ∼ 0.55. But when p gradually deviates from 0.5, collapse occurs."
The whitening function is the case where $p=0.5$.

In section 3.3, the iterative normalization has also been explored by [40] in https://github.com/PatrickHua/FeatureDecorrelationSSL/blob/main/models/utils/iterative_normalization.py

The major contribution of this paper in my view is to propose a trace loss in iterative normalization to avoid dimension collapse. Yet, as the quantitative comparison to [20,40] is missing, it is hard to evaluate the contribution.




**Questions:**

Please refer to the weakness

**Limitations:**

I think the authors well address the limitations.

---

> ### Author Rebuttal · Authors · 2023-08-07
>
> ## Response to Reviewer TM8b
>
> We thank the reviewer for the encouraging and insightful comments. Please find our responses to specific concerns below.
>
> **Concern 1:** Comparing methods in the Experiment section: As the proposed method is claimed to be the general form of whitening function in SSL, I am wondering why the authors did not put the performance comparison against whitening loss [20, 40]
>
> **Response:** Thanks sincerely for your constructive suggestions. The reason why we did not provide results for shuffled-DBN[20] and CW-RGP [40] is that the results of all baselines in Table 1 (Evaluation on ImageNet) of our paper are nearly pre-trained for 800 or 1000 epochs, while the official results of shuffled-DBN [20] and CW-RGP [40] we can find in corresponding papers [20,40] are only pre-trained up to 200 epochs at most. We think it is unfair to directly compare [20,40] to other baselines in Table 1 that have long-term training. We thus did not put the performance comparison against [20,40].
> Following your insightful suggestions, we also conduct additional experiments to compare against [20,40] under the same experimental setup as ours. We first show the results on CIFAR-10/100 and ImageNet-100 in the table below, following the same setup in Table 2 of our paper. We can see that INTL outperforms shuffled-DBN and CW-RGP consistently over all datasets.
>
> | Method            |              CIFAR-10              |              CIFAR-100               |            ImageNet-100            |
> | :---------------- | :--------------------------------: | :----------------------------------: | :--------------------------------: |
> |                   |   top-1 &nbsp; 5-nn &nbsp; top-5   |    top-1 &nbsp; 5-nn &nbsp; top-5    |   top-1 &nbsp; 5-nn &nbsp; top-5   |
> | Shuffled-DBN [20] |  91.17 &nbsp; 88.95 &nbsp; 99.62   |   66.81 &nbsp; 57.27 &nbsp; 90.78    |   75.27&nbsp; 67.21&nbsp; 93.12    |
> | CW-RGP [40]       |  92.03 &nbsp; 89.67 &nbsp; 99.73   |   67.78 &nbsp; 58.24  &nbsp; 90.65   |   76.96 &nbsp;68.46 &nbsp; 93.76   |
> | INTL (ours)       | **92.60 &nbsp; 90.03&nbsp; 99.80** | **70.88 &nbsp; 61.90 &nbsp;  92.13** | **81.68 &nbsp;73.46 &nbsp; 95.42** |
>
> For the experiments on ImageNet, we only compare INTL to shuffled-DBN[20] and CW-RGP [40] under 200 epochs pre-training due to time limit of the rebuttal period, following the same experimental setup as in Table 1 of INTL. The results are shown in the table below. We can see INTL also outperforms shuffled-DBN and CW-RGP consistently. The schedule to train shuffled-DBN and CW-RGP for 800 epochs are currently in the pipeline. We would like to add these results in the revised version.
>
> | Method            |        ImageNet         |
> | :---------------- | :---------------------: |
> |                   |   top-1 &nbsp;  top-5   |
> | Shuffled-DBN [20] |   65.18 &nbsp;  85.32   |
> | CW-RGP [40]       |   69.72 &nbsp;  88.92   |
> | INTL (ours)       | **71.10  &nbsp; 90.61** |
>
>
> **Concern 2:** Concerns about novelty. In my view, section 3.2 Spectral Transformation proves the effectiveness of Whitening. The general form of ST is $g(\lambda)=\lambda^{-p}$.
>
> **Response:** We first clarify the relationship among whitening, Spectral Transformation (ST) and power function ($g(\lambda)=\lambda^{-p}, p \in \mathbb R$) for ST.  In our Spectral Transformation framework,  $g(\lambda)$ can be any function which finally maps the eigenvalue  $\lambda$ to $\lambda g^2(\lambda)$. Therefore, both whitening ($g(\lambda)=\lambda^{-1/2}$) and power function ($g(\lambda)=\lambda^{-p}, p \in \mathbb R$) are special cases of ST.  Note that  $g(\lambda)$ of IterNorm with T iteration number is $f_T(\frac{\lambda}{tr(\Sigma)}) / \sqrt{ tr(\Sigma)}$(Line 218, and $f_T(\cdot) $is defined in Line 216), which is not a power function, but an ST by definition.
> We agree that ST proves the effectiveness of Whitening ($p=0.5$) in section 3.2. Meanwhile, we also want to show that there exist other functions for ST except whitening that can also work to avoid collapse in section 3.2 (e.g., the power function for ST with $p = 0.45$ or $0.55$ in Figure 2). This result drives us to believe that other effective ST can be designed/found to avoid dimensional collapse. Perhaps our unclear description caused your misunderstanding, and we will further refine this section.
>
> **Concern 3:** In section 3.3, the iterative normalization has also been explored by [40] in the github.
>
> **Response:** We thank the reviewer for pointing out that iterative normalization (IterNorm) algorithm has been listed in the github. By checking this github, we believe it is probably the released codebase of the Shuffled-DBN paper [20] (rather than the CW-RGP paper [40]). However, we do not find any description or experiment relating IterNorm in the Shuffled-DBN paper [20]. The paper [20] uses the original ZCA batch whitening algorithm for Shuffled-DBN in its experiments. Note that our paper shows that IterNorm suffers severe dimensional collapse and mostly fails to train the model in SSL. That is why, we guess, Shuffled-DBN paper [20] uses ZCA batch whitening algorithm rather than IterNorm, even though IterNorm is in its github. Moreover, we propose IterNorm with trace loss (INTL), a solution to address the failure of IterNorm, and we theoretically prove that INTL can avoid collapse and modulate the spectrum of embedding towards an equal-eigenvalue distribution during the course of optimization.

---

### Author Rebuttal · Authors · 2023-08-09

## Global Response to Reviewers

We thank all the reviewers for their detailed and constructive comments. We briefly highlight the merits recognized by reviewers as follows:
1. Great representation, for example, “the paper is well-organized with a clear story and justification” (TM8b), “the presentation is great” (QdWo), and “The presentation of the paper is good, with a clear elaboration of the motivation and contribution of the proposed method” (Fnki).
2. Theoretical contribution, for example, “this paper conducts rigorous theoretical analysis” (SEtq), “the paper proposes an elegant method with theoretical guarantees” (Fnki), and “the paper prove the previous whitening function is a special case of spectrum transformation (ST)” (TM8b).
3. Empirical success, for example, “The paper shows INTL reaches SOTA performance on image classification and object detection tasks” (Ckne), “State-of-the-art SSL performances using ResNet50 were achieved” (tQK6), and “the proposed method is evaluated on several SSL-evaluation benchmarks, and significantly surpasses the previous state-of-the-art methods” (Fnki)

We respond to each reviewer in the separate local ‘Rebuttal’ section. We also conduct additional experiments using vision transformer (ViT) backbones, following the suggestion of Reviewer Fnki. The results are shown in Figure 1 of the ‘pdf’ file of global ‘Rebuttal’ section.

---

### Decision · Program_Chairs · 2023-09-21

**Decision:**

Reject

**Comment:**

The paper addresses an important problem in self-supervised contrastive learning on feature collapse via spectrum modulation.
Most of the reviewers find the paper well-presented, the method is very intuitive and works well with small batches, the effectiveness of the proposed method is well demonstrated on large datasets.

However, as Reviewer Ckne raised, this paper resembles quite a few similarities to recent AAAI work: "Spectral Feature Augmentation for Graph Contrastive Learning and Beyond", but the work has not been carefully cited nor discussed in the submission.
In the rebuttal, the authors have not fully addressed this concern and the novelty over the previous AAAI work.

Given the concerns, we regret to reject the paper and recommend the authors to fully discuss and acknowledge the related AAAI work before future submission